# Organ-specific immune responses are strain-dependent in a mouse model of *Cryptococcus neoformans* brain infection

Man Shun Fu,[1] Lorna George,[1] Daisy Harris-Bosancic,[1] Tahrim Hussain,[1] Erin Clipston,[1] Pui Mun Emily Chan,[1] Lozan Sheriff,[1] David Lecky,[1] Kazuyoshi Kawakami,[2] Rebecca A. Drummond[1]

**ABSTRACT**  Cryptococcal meningitis remains the top cause of death from an invasive fungal infection in humans, responsible for ~100,000 deaths annually of vulnerable patients with underlying immune deficiencies. Animal models of cryptococcal meningitis are important for understanding the immune parameters that correlate with protection. However, modeling this infection in mice is challenging. There is wide variability in infection routes, doses, and mouse background used in the field, which makes understanding phenotypes of mutants and immune interventions difficult to broadly apply. Our intention was to create an observational data set for the field on how *Cryptococcus neoformans* strain influences analysis of organ-specific immune responses in an intravenous mouse model of cryptococcal meningitis, focusing on impact of the fungal strain while keeping mouse genetic background (C57BL/6J) and infection route constant. We quantified myeloid and lymphoid cell recruitment and fungal-specific CD4 T-cell activation, correlating these results with fungal burdens in mice infected with the commonly used reference strain H99 or with two recently isolated clinical strains that were the same molecular type (VNI) or an unrelated type (VNB). We also analyzed how dose used in murine infection models affected brain immune responses during *C. neoformans* infection. Our work reveals intriguing patterns of organ-specific immunity that are dependent on *C. neoformans* strain but not always explained by virulence potential, raising important future questions for the field regarding the impact of *C. neoformans* strain on cellular immune responses in experimental animal models.

**IMPORTANCE**  Cryptococcal meningitis is a fungal infection that causes a wide variation of clinical disease in patients. This variation is thought to be partly due to the diversity of fungal strains that cause the infection. In this work, we have provided an in-depth analysis of immune responses to different clinical isolates of the fungus using mouse models of the infection. Our work reveals intriguing patterns of organ-specific immunity that are dependent on *C. neoformans* strain but not always explained by virulence potential, raising important future questions for the field regarding the impact of *C. neoformans* strain on cellular immune responses in experimental animal models.

**KEYWORDS**  *Cryptococcus neoformans*, neuroimmunology, central nervous system infections, microglia, CD4 T-cell, strain variation

T he fungus *Cryptococcus neoformans* is the main causative agent of fungal meningitis in humans (1). *C. neoformans* is prevalent in the environment, and exposure is typically via inhalation of spores. In patients with compromised cellular immunity, particularly CD4 T-cells, *C. neoformans* spores may germinate into yeast and establish infection. *C. neoformans* infections often have central nervous system (CNS) involvement (cryptococcal meningitis), which is associated with a high mortality rate and neurological impairment in survivors (2). Protection against this infection requires CD4 T-cells, which

**Peer Reviewer** Floyd L. Wormley, Texas Christian University, Fort Worth, Texas, USA

Address correspondence to Rebecca A. Drummond, r.drummond@bham.ac.uk.

The authors declare no conflict of interest.

See the funding table on p. 18.

instruct B-cells to make anti-fungal antibodies that are required for effective uptake by myeloid cells. CD4 T-cells also produce pro-inflammatory cytokine IFNγ, which provides killing signals to myeloid cells and prevents intracellular survival of the fungus within macrophage phagosomes (3).

*C. neoformans* was first thought to be one species but is now recognized to encompass distinct species and several molecular types. There are two species complexes, *C. neoformans* (the focus of this study) and *C. gattii* (4). *C. neoformans* complex typically causes CNS disease in immunocompromised individuals, while *C. gattii* infections are more geographically restricted and develop in patients with unclear immune status. Each complex is split into serotypes based on antigenic differences within the capsule of the fungus, with serotypes A and D within the *C. neoformans* complex, and B and C within the *C. gattii* complex (4). The majority of human infections are caused by serotype A strains (referred to as *C. neoformans*) (5). Within the serotype A grouping, there are several molecular types which have wide diversity in geographical distribution, genetic,s and clinical presentation. Molecular type VNI and VNII strains are found globally, but VNII strains are not typically found in clinical samples (5). VNB strains are more common in South Africa and South America, and are associated with a poor patient outcome (5).

To better understand pathogenesis and mechanisms of disease that operate during cryptococcal meningitis, investigators have developed several animal models of *C. neoformans* infection in rabbits (6), rats (7), zebrafish (8), and mice (9). Mouse models are more common in immunology studies since they have both an innate and adaptive immune system that closely resembles the human immune system, and there are significantly more transgenic lines and molecular tools to study murine immunity than are available for other animal models. Multiple infection routes have been implemented in mice to model *C. neoformans* infection. The most commonly used are intranasal and intratracheal routes, where yeast is delivered into the nasal passage and/or airways to initiate pulmonary infection. Even in immunocompetent mice, virulent strains of *C. neoformans* escape the lung without further intervention and disseminate to other organs, primarily the CNS (9). These models have been used to examine how pulmonary immunity develops and the mechanisms or drug treatments required to prevent dissemination to the brain. However, dissemination to the CNS in these models is often variable, and it can be difficult to separate mechanisms of pulmonary immunity with those of CNS-specific immunity. Researchers have therefore used more direct routes to establish brain infection that bypass pulmonary immunity, allowing a targeted understanding of CNS disease and brain-specific immune responses. These include the intravenous and intracerebral injection models, although the latter route is less common due to the expertise required to perform such injections.

In addition to the variety of infection routes used in the field, there is also variation in the *C. neoformans* strains used. The most commonly used strain to model cryptococcal infection in mice is H99, a serotype A strain (molecular type VNI) that was first isolated in the 1970s at Duke University from a lymphoma patient (10). It is highly virulent in mice and is useful for modeling the inflammatory responses that develop to this fungal infection *in vivo*. When injected intravenously, H99 drives an acute meningitis in wild-type mice within 1 week (11). This strain is also a good model for examining mechanisms of intracellular fungal survival in macrophages, since it exhibits good survival rates in the phagosomes of cultured macrophages and *in vivo* within myeloid populations (11, 12). Due to these useful traits, several knock-out mutant libraries and fluorescent reporter strains have been generated on this strain background. However, there are reports of genetic drift in H99 isolates used in different laboratories (10), and concerns have been raised as to how representative this strain is, particularly as other clinical isolates tested in mice appear to be significantly less virulent (13). Another strain that is popular in the field is *C. deneoformans* 52D, a serotype D strain. When injected intravenously, 52D exhibits a strong tropism for the CNS and results in a chronic infection that resembles clinical features of immune reconstitution inflammatory syndrome (IRIS), which is a major driver of neurological sequelae in patients (14). Infections with 52D are

often used to model the immune events that contribute towards pathological inflammation during cryptococcal meningitis. Despite the advantages that strains such as H99 and 52D provide, recent studies have indicated that a wider range of clinical isolates need to be tested to identify clinically relevant virulence factors, particularly those mediating latent infections (15).

In the current study, we aimed to comprehensively characterize organ-specific cellular immune responses in wild-type C57BL/6 mice infected with *C. neoformans* and analyze the impact of fungal strain background using a limited number of strains, with the goal of providing the community with data sets that can be used to select reference strains to model an immune response of interest. We have primarily focused on immune responses occurring within the brain for this study, as these are the most poorly understood and have therefore used the intravenous infection route to specifically study CNS-localized immune responses.

## RESULTS

### Organ-specific virulence of *C. neoformans* strains

*C. neoformans* may invade multiple organs during experimental infection, particularly following intravenous inoculation. The intravenous infection model, therefore, gives a broad analysis of organ-specific virulence potential (by measuring fungal burdens) across multiple tissue sites in the same animal. Prior to performing quantitative analysis of cellular immune responses in mice infected with different *C. neoformans* strains, we first assessed the virulence of recently isolated *C. neoformans* strains in the brain, lung, liver, and spleen. We used the reference strain H99, which is well characterized in mouse models, and compared it with two strains that were isolated from clinical samples taken from patients in Zimbabwe (16). We chose Zc1, which is distantly related to H99 but is the same molecular type (VNI) and exhibits a significant impairment in its ability to form Titan cells *in vitro* and *in vivo* (17), a key trait for *C. neoformans* virulence. We also chose Zc15, a VNB strain that is a molecular type associated with poor clinical outcome in humans (5). We infected healthy C57BL/6 mice with each of the three strains intravenously and assessed survival and weight loss over the first two weeks of infection. As expected, we found that H99 was the most virulent strain, with significantly greater mortality rate compared with both Zc1 and Zc15 (Fig. 1A). There was 100% lethality following infection with either H99 and Zc1, whereas ~40% of mice survived infection with *C. neoformans* Zc15 at 2 weeks post-infection. In the first 9 days of infection (when mice in all groups were still surviving), we found significant loss of weight in mice infected with H99, whereas mice in other groups had not lost weight (Fig. 1B).

Next, we measured fungal burdens in different organs at day seven post-infection. In line with our survival data, mice infected with H99 had significantly higher fungal burdens in every organ tested compared with the other two strains, with the highest levels observed in the brain (Fig. 1C). In the brain, there was significantly less fungal burdens in mice infected with Zc1 and Zc15, with Zc15 exhibiting ~50-fold less fungal burdens than Zc1 and ~120-fold less than H99 (Fig. 1C). Interestingly, this pattern was not as evident in other organs. In the lung and liver, Zc1 and Zc15 had similar fungal burdens, and both were less than H99 (Fig. 1C). In contrast, H99 and Zc1 infected the spleen to a similar degree, while Zc15 had the least fungal burdens in this tissue (Fig. 1C). All strains had a strong propensity to grow in the brain, with 10–100-fold higher growth in brain than other tissues. We recovered similar burdens of Zc15 from the liver and spleen, while H99 and Zc1 had greater fungal burdens in the liver than in the spleen (Fig. 1C). Taken together, these data show organ-specific patterns of virulence that are dependent on fungal strain.

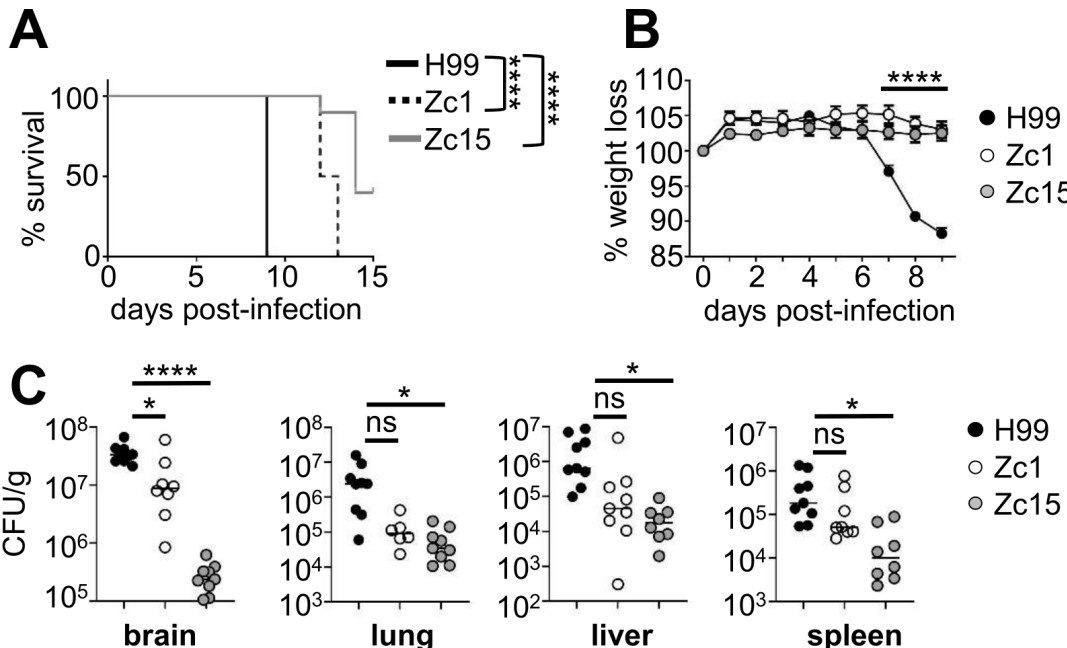

**FIG 1** *C. neoformans* strain variation drives organ-specific differences in fungal burden and virulence. (A) Wild-type C57BL/6J mice were infected intravenously with $2 \times 10^4$ CFU of *C. neoformans* H99 (*n* = 10 mice), Zc1 (*n* = 10 mice) or Zc15 (*n* = 10 mice). Animals were monitored for 15 days post-infection and euthanized when humane endpoints were met. Data are pooled from two independent experiments and analyzed by Log-rank Mantel-Cox test. ****$P$ < 0.0001. (B) Infected mice (H99, *n* = 10 mice; Zc1, *n* = 10 mice; Zc15, *n* = 10 mice) were weighed daily and weight expressed as % change to starting weight on day of infection. Data are pooled from two independent experiments and analyzed by two-way ANOVA. ****$P$ < 0.0001 (H99 vs Zc1; H99 vs Zc15). (C) Fungal burdens of the indicated organs were measured on day 7 post-infection in mice infected with *C. neoformans* H99 (*N* = 9 mice), Zc1 (*N* = 9 mice), or Zc15 (*N* = 8 mice). Each point represents an individual animal. Data are pooled from three independent experiments and analyzed by one-way ANOVA with Tukey multiple comparison test. *$P$ < 0.05, ****$P$ < 0.0001. ns = not significant.

## Brain invasion pattern and cryptococcoma size are influenced by *C. neoformans* strain

We next examined the pathological features in the brains of mice infected with *C. neoformans* H99, Zc1, and Zc15 using histology. In the brain, *C. neoformans* invades the brain parenchyma and forms large lesions called cryptococcomas and is also typically found in the perivascular spaces where it proliferates and contributes to inflammation in this tissue (18). Consistent with the fungal burdens quantified from whole brain (Fig. 1C), we observed less cryptococcal cells in lesions of brains infected with Zc15 in comparison with H99 (Fig. 2A). The size of the yeast within brain lesions was similar for all fungal strains, and we did not observe Titan cells in the brain in any sections analyzed (Fig. 2A). The brain lesions caused by Zc15 infection were significantly smaller compared with those caused by H99 and Zc1 (Fig. 2B). However, there was little difference in the total number of lesions we observed in the brain between strains (Fig. 2C). Regardless of the *C. neoformans* strain used, the infiltration of inflammatory cells was lacking around the parenchymal lesions containing fungal cells (Fig. 2A). In contrast, blood vessels were surrounded by infiltrating perivascular leukocytes in response to H99 infection, whereas only small numbers of immune cells were observed in the perivascular space when mice were infected with *C. neoformans* Zc1 or Zc15 (Fig. 2D). There was no consistent pattern between the fungal strains in which brain regions the fungal cells localized to (Fig. S1). Taken together, histological analysis revealed strain-dependent variation in parenchymal lesion size and perivascular inflammation, which largely mirrors the differences observed in levels of live fungus recovered from brain tissue.

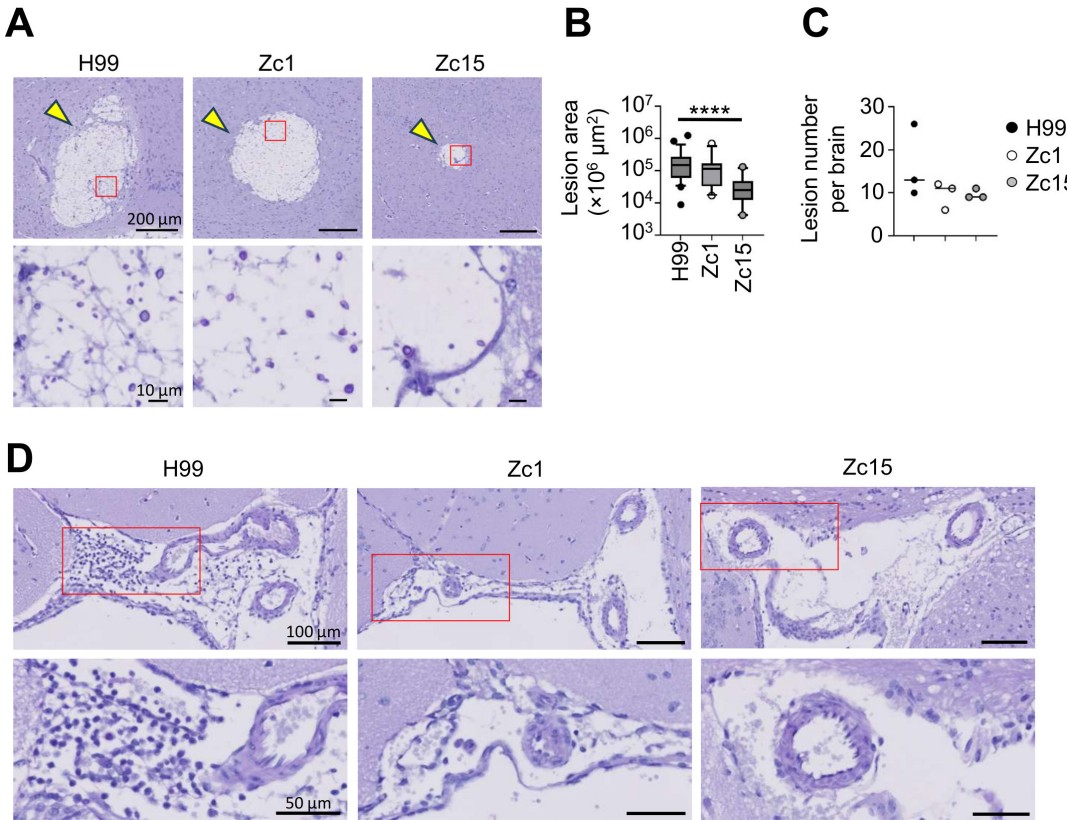

**FIG 2** *C. neoformans* strain variation drives differences in brain lesion size and inflammatory infiltrate. (A) Example PAS-stained histology of infection sites in brain sections of mice infected with *C. neoformans* H99, Zc1, or Zc15 at day 7 post-infection. Yellow arrow indicates lesions in the brain. Enlarged views of the red boxed areas in the upper panels are shown in the lower panels to better visualize cryptococcal cells within the lesions. Three mice per strain were analyzed to confirm the observations. (B) The size of all individual lesions was quantitated across three mouse brains for each fungal strain. The central line denotes the median value (50th percentile), while the box contains the 25th to 75th percentiles of the data set. The whiskers mark the 5th and 95th percentiles, and values beyond these upper and lower bounds are considered outliers. (C) The number of lesions per mouse brain was counted across three mouse brains for each fungal strain. Each point represents an individual animal. For panels B and C, data were analyzed by one-way ANOVA with Tukey correction. ****$P < 0.0001$. (D) Representative histology of brain perivascular space (PVS). Histology was performed 7 days post-infection, stained with PAS. Enlarged views of the red boxed areas in the upper panels are shown in the lower panels to better visualize immune cell infiltration.

## Influence of brain fungal burden on inflammatory cell recruitment is cell type and strain-specific

We next performed broad immune phenotyping of tissue-resident macrophage populations and recruited inflammatory cells in the brains of mice infected with *C. neoformans* H99, Zc1, and Zc15 to understand whether the differences in fungal burdens we observed between these strains translated into quantitative differences in cellular immune responses. In the brain, we found a similar number of microglia, the tissue-resident macrophages of the CNS, in all mice regardless of *C. neoformans* strain (Fig. 3A). We found similar numbers of macrophages in mice infected with H99 and Zc1, with significantly reduced numbers of macrophages in Zc15-infected mice (Fig. 3A). In contrast, inflammatory Ly6C^hi monocytes, which are recruited to the brain upon infection, were significantly reduced in mice infected with Zc1 and Zc15, compared with H99 (Fig. 3A). Neutrophils, which are also recruited upon infection, had variable numbers across the groups, particularly Zc1, and there was no significant difference between H99 and Zc1 (Fig. 3A). Zc15-infected mice had significantly reduced neutrophil recruitment to the brain, as we had observed for Ly6C^hi monocytes (Fig. 3A). Since the pattern in numbers of some of the immune populations (microglia, macrophages, and neutrophils) did not mirror the difference we observed in fungal brain burdens between

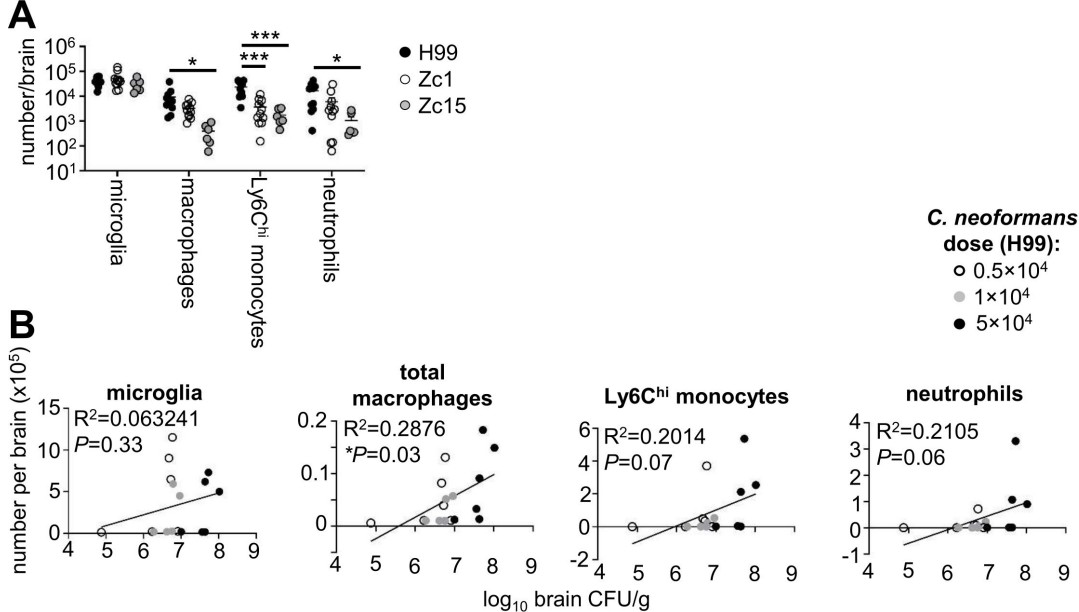

**FIG 3** Quantification of immune cell populations in brains of mice infected with *C. neoformans* H99, Zc1, and Zc15. (A) Mice infected with *C. neoformans* H99 ($n$ = 11 mice), Zc1 ($n$ = 11 mice), and Zc15 ($n$ = 6 mice) were analyzed at day 7 post-infection. Total numbers of indicated myeloid cell populations were quantified using flow cytometry. Data are pooled from two independent experiments and analyzed by two-way ANOVA. *$P < 0.05$, ***$P < 0.005$. (B) Mice were infected with different doses of *C. neoformans* H99 ($n$ = 6 mice per dose), and indicated myeloid cell populations were quantified using flow cytometry and correlated with fungal brain burden (matched to the same animal). Data are pooled from two independent experiments and analyzed by simple linear regression.

H99 and Zc1 (Fig. 1), we next explored how fungal burden in the brain influenced leukocyte numbers to determine whether it was fair to assume that higher fungal burdens would equate to more immune cells. For that, we infected mice with different doses of H99 and correlated immune cells in the brain with fungal burdens measured in the same animal (Fig. 3B). These results showed that there was no correlation between infection dose, fungal brain burden, and numbers of tissue-resident microglia, indicating that numbers of these cells are not influenced by fungal brain burden in this acute infection model (Fig. 3B). In contrast, there was a significant correlation between fungal brain burdens and numbers of macrophages, and a positive trend between neutrophils and Ly6C$^{hi}$ monocytes that did not quite reach statistical significance, indicating that inflammatory responses measured by proliferation and/or recruitment of these cells are likely influenced by fungal brain burden (Fig. 3B). It is therefore likely that the reduction in macrophages, monocytes, and neutrophils we observed in the brains of Zc15-infected mice (Fig. 3A) is the result of reduced fungal brain burdens we observed with this strain (Fig. 1C). However, we saw no difference in neutrophil or macrophage numbers in brains infected with Zc1 (Fig. 3A), despite significantly reduced fungal burden compared with H99 (Fig. 1C), indicating that there are additional strain-dependent effects that influence recruitment of these populations. Collectively, these experiments revealed positive correlations between fungal brain burden and recruitment of most immune cell types that were unaffected by *C. neoformans* strain variation, with resident myeloid cells unaffected by brain burden and fungal strain.

## Cryptococci congregate near blood vessels in the liver

Liver dysfunction and advanced liver disease are independent risk factors for several fungal infections, including those caused by *C. neoformans* (19). Resident liver macrophages, called Kupffer cells, are important for controlling disseminated cryptococcal infections by capturing yeast in a C3-dependent manner (20). We used histology to examine where *C. neoformans* localizes within the liver and analyzed the size and composition of infection sites to determine if fungal strain variation affected these

parameters. We found inflammatory lesions in the livers of infected mice were typically found in the vicinity of a blood vessel (Fig. 4A), in line with the use of the intravenous infection route in this model. While more than half of these lesions (65%) contained visible fungal cells in mice infected with H99, this was lower in Zc1 and Zc15 infected animals (45% and 40%, respectively) (Fig. 4A and B). Lesion size and number were also affected by fungal strain, which was largely influenced by fungal burdens measured in the liver. Lesion sizes were similar between H99 and Zc1, regardless of whether lesions contained fungal cells or not (Fig. 4C). However, there were significantly fewer lesions in Zc1-infected liver compared with H99 (Fig. 4D). Zc15-infected liver had the smallest lesion areas, particularly if they did not contain visible fungal cells, and contained a similar number of total lesions to Zc1-infected livers. Taken together, these data show qualitative differences in infection lesions found in the liver that are *C. neoformans* strain dependent.

## Lung resident macrophage populations are influenced by *C. neoformans* strain independent of lung fungal burden

*C. neoformans* infections are thought to originate in the lungs following inhalation of spores from the environment (21). Here, fungal cells closely interact with several populations of resident lung macrophages, which demonstrate differing responses to fungal immune modulators and intracellular infection (22, 23). To examine how *C. neoformans* strain variation impacted antifungal immunity and pathology in the lung, we examined lung inflammatory scores using histology and quantified populations of resident and infiltrating immune cells by flow cytometry in mice intravenously infected with H99, Zc1, and Zc15. As observed in the liver, intravenous infection resulted

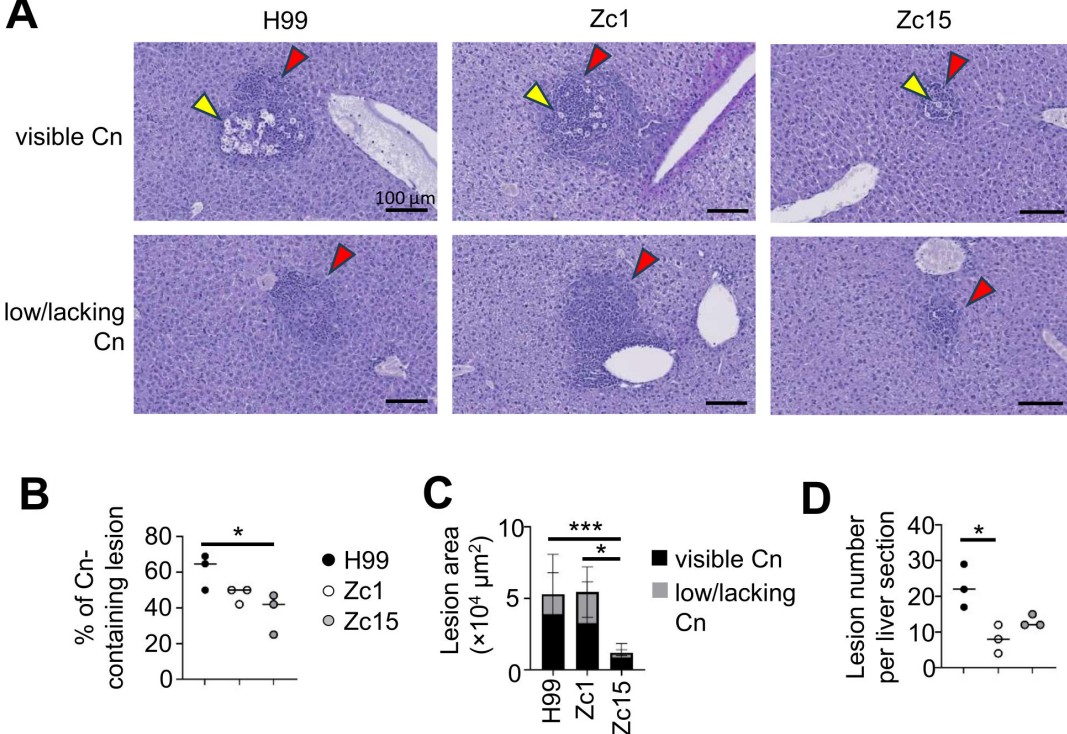

**FIG 4** *C. neoformans* strain variations in liver infection sites. (A) Representative PAS-stained liver sections from infected mice at day 7 post-infection. Example sections are taken from the same animal. Upper panels show sites containing cryptococcal cells (yellow arrows) surrounded by inflammatory cells (red arrows), while the lower panels show inflammation in the absence of *C. neoformans* at those locations. Three mice per fungal strain were analyzed to confirm the observations. (B) The percentage of Cn-containing lesions was compared among livers infected with H99, Zc1, and Zc15. (C) The size of all individual lesions was measured across three mouse livers for each fungal strain, comparing lesions with and without cryptococcal cells. (D) The number of lesions per mouse liver was counted across three mice for each fungal strain. For panels B–D, data were analyzed by one-way ANOVA with Tukey correction. *$P < 0.05$, ***$P < 0.001$.

in cryptococcal cells localizing close to blood vessels but were also observed near bronchioles. We were only able to consistently observe these infection sites in mice infected with *C. neoformans* H99, as there were significantly fewer yeast cells observed in lung sections from mice infected with Zc1 and Zc15 (Fig. 5A). We observed several instances of Titan cell formation in the lungs of H99 infected animals, aligning with previously findings which indicated Titan cell formation is likely a lung-specific phenotype and occurs at a much lower rate in the Zc1 strain (17).

In mice infected with H99 and Zc1, cryptococcal cells were surrounded by infiltration of inflammatory cells, whereas this was not observed in Zc15-infected lung (Fig. 5A). However, alveolar thickening and increasing bronchial mucosal hyperplasia and hypertrophy were present for all *C. neoformans* strains (Fig. 5A). These inflammatory features, along with PAS-positive cells in bronchioles, were also found in some areas where cryptococcal cells are absent in all three infection groups (Fig. S2). Quantitative analysis of histology observations showed that alveolar thickening was more pronounced in the H99-infected lung and less in the Zc1 and Zc15-infected lungs (Fig. 5B), mirroring culturable fungal burdens from lung with these strains (Fig. 1C). We found no difference in peribronchial infiltration between the strains (Fig. 5C), and there was no clear difference in the number of PAS-positive cells in the bronchioles between the strains (Fig. 5D).

To examine immune cell populations in the lung in more detail, we used flow cytometry to quantify the numbers of tissue-resident and recruited immune cells in the lungs of mice infected with H99, Zc1, and Zc15. In the lung, there are two major populations of resident macrophages. Alveolar macrophages are found in the airways, and interstitial macrophages reside in the tissue. Interstitial macrophages can be further divided into MHCII$^{hi}$ and MHCII$^{lo}$ subsets, which have different spatial localizations and functions (25). Interestingly, we found that the number of all three subsets of macrophages was significantly higher in the Zc1-infected lung compared with the other two strains (Fig. 5E). Alveolar macrophage numbers did not differ between H99 and Zc15, while we observed a trend of less interstitial macrophages in the Zc15-infected lung compared to H99 (Fig. 5E). In contrast to the brain, numbers of Ly6C$^{hi}$ monocytes did not differ in the lung between any of the strains (Fig. 5E). We observed a significant reduction in neutrophils in the Zc15-infected lung compared with H99, which had comparable numbers with Zc1 (Fig. 5E). Eosinophils, which increase in number during *C. neoformans* lung infection, trended higher in the Zc1-infected lung but otherwise were similar between the three strains (Fig. 5E). Collectively, these data show that numbers of lung leukocytes do not necessarily follow similar patterns as observed with lung fungal burdens and demonstrate strain-specific effects on the population dynamics of lung leukocytes, particularly of lung-resident macrophage populations.

## *C. neoformans* strain variation significantly influences granulocyte recruitment to the lung following intranasal infection

Although we had observed intriguing differences in lung-resident macrophages between *C. neoformans* strains that did not align with fungal burden, these differences were detected using an intravenous infection model, which is considered a better model for meningitis and brain infection, as the lung does not become heavily infected using this infection route and granulomas are not typically found. Instead, lung pathology that more accurately mimics clinical infection features is best observed using an inhalation infection model. We therefore performed intranasal infections with *C. neoformans* H99, Zc1, and Zc15 and determined whether these strains exhibited similar virulence profiles using this infection route, and whether the differences we observed in lung macrophages between the strains were consistent using the intranasal infection route. We found that the *C. neoformans* strains broadly had similar virulence levels in the intranasal model as seen with the intravenous model; the highest lung fungal burdens were observed in the H99-infected lung at 3 weeks post-infection, while Zc15 had the lowest (Fig. 5F). Similar trends were seen in the brain, spleen, and liver (Fig. 5F).

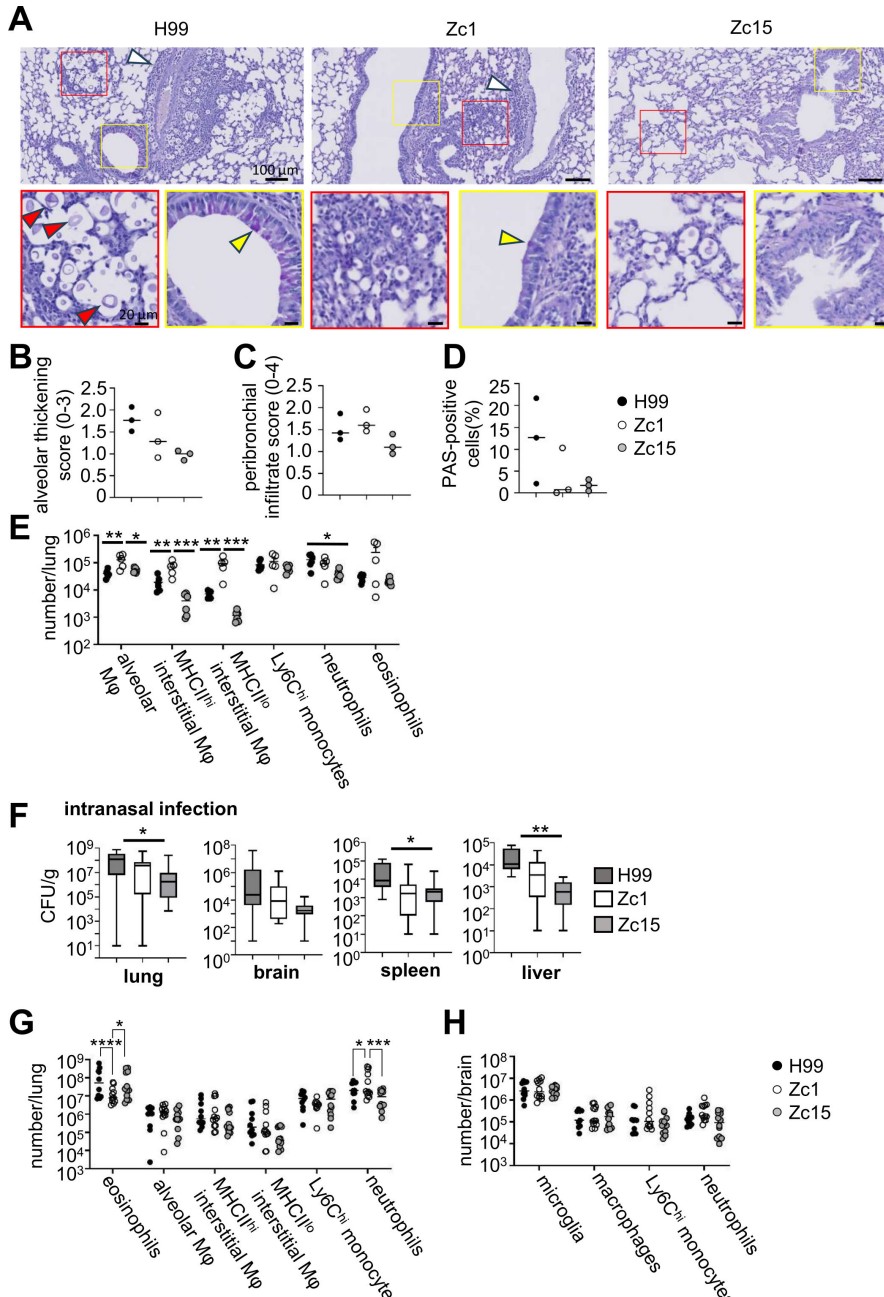

**FIG 5** Lung pathology and local immune populations are influenced by *C. neoformans* strain and infection route. (A) Representative PAS-stained histology sections from mice infected with *C. neoformans* H99, Zc1, and Zc15 at day 7 post-infection. White arrows indicate inflammatory cell infiltration. Enlarged views of the red boxed areas in the upper panels are shown in the lower panels to better visualize *C. neoformans* cells. Red arrow indicates titan cells. Yellow boxes (upper panels) denote area of accompanying enlarged images (lower panels), which show bronchioles. Yellow arrow indicates PAS-positive cells. Three mice per fungal strain were analyzed to confirm the observations. (B) Alveolar involvement/thickening was blindly scored (60–80 images per mice, three mice for each strain) on a scale from 0 to 3 based on published criteria (24). (C) Peribronchial infiltrate was scored (0–4) in all individual bronchioles across three mice for each strain based on published criteria (24). (D) Quantification of PAS-positive cells in bronchioles of three mice for each strain. The number of PAS-positive cells was counted in a total of 100 epithelial cells in each bronchiole. (E) Total number of indicated myeloid populations in the lung of mice infected with *C. neoformans* H99 ($N = 6$ mice), Zc1 ($N = 5$-6 mice), and Zc15 ($n = 5$-6 mice). Data pooled from two independent experiments and analyzed by two-way ANOVA. *$P<0.05$, **$P<0.01$, ***$P<0.005$. (F) Mice were intranasally infected with $2\times10^5$ *C. neoformans* H99 ($N = 15$ mice), Zc1 ($N = 16$ mice) and Zc15 ($n = 16$ mice) and analyzed at day

Fig 5 (Continued)

21 post-infection for tissue fungal burdens in the indicated organs. Data is pooled from three independent experiments and analyzed by one-way ANOVA. *$P < 0.05$, **$P < 0.01$. (G) Total number of indicated myeloid populations in the lung and (H) brain of mice intranasally infected with *C. neoformans* H99 ($n = 10$ mice), Zc1 ($n = 12$ mice), and Zc15 ($n = 12$ mice). Data pooled from two independent experiments and analyzed by two-way ANOVA. *$P < 0.05$, ***$P < 0.005$, ****$P < 0.001$.

Next, we quantified immune cell populations in the lungs of intranasally infected mice on day 21 post-infection. We found no differences between the strains in the numbers of lung-resident macrophages or monocytes (Fig. 5G), in contrast to our earlier observations in the intravenous infection model (Fig. 5E). However, we did observe significant differences in the recruitment of granulocyte populations to the Zc1-infected lung (Fig. 5G). There were significantly more neutrophils, but fewer eosinophils, in the Zc1-infected lung compared with H99- and Zc15-infected lungs, which were similar to one another (Fig. 5G). In the brain, we found no difference between *C. neoformans* strains in any of the myeloid populations quantified following intranasal infection (Fig. 5H).

Collectively, these data show that while strain-dependent difference ins fungal virulence levels were largely similar between different infection routes, differences in immune cell populations were specific to the infection route used.

## CD4 T-cell and NK cell recruitment positively correlates with fungal brain burden

We next examined how *C. neoformans* strain variation and fungal brain burden affected recruitment of lymphocyte populations during infection. Lymphocytes are required for protection against *C. neoformans* infections in humans, particularly CD4 T-cells and B-cells, which provide IFNγ-mediated killing signals to myeloid cells and produce antibody for effective opsonization and phagocytosis, respectively (3). We quantified lymphocyte numbers in the brains of mice infected with H99, Zc1, and Zc15 using flow cytometry. We used the intravenous model of infection for these experiments, since lymphocyte recruitment to the brain following intranasal infection is more variable due to the lower brain burdens achieved with this infection route (compare Fig. 1C with Fig. 5F). We found no difference in the number of CD8 T-cells or B-cells in the brains of infected mice between any of the strains (Fig. 6A). Numbers of NK cells largely mirrored the fungal brain burden differences between these strains, with greater numbers in the H99-infected brain, followed by Zc1 and then Zc15 (Fig. 6A). We found that H99 infection induced the highest number of brain-infiltrating CD4 T-cells during infection (Fig. 6A). Numbers of CD4 T-cells in the brains of mice infected with Zc1 and Zc15 were significantly less than H99 and were similar to each other, with Zc1 exhibiting more variability between mice (Fig. 6A). CD44 expression, a marker of T-cell activation, was largely similar by CD4 T-cells regardless of *C. neoformans* strain used although there was a slight trend of reduced CD44 expression during infection with Zc1 and Zc15 (Fig. 6B), which largely mimicked the pattern seen in brain fungal burdens (Fig. 1C). Indeed, mice infected with different doses of H99 exhibited a correlation between brain fungal burden and number of recruited CD4 T-cells, particularly for effector memory and activated (CD44[+]) CD4 T-cell subsets (Fig. 6C). Next, we analyzed T-cell polarization in the brain using intracellular flow cytometry. We found that the frequency of CD4 T-cells that were producing IFNγ, IL-13, or IL-17 was largely similar between the three *C. neoformans* strains (Fig. 6D). However, we found a significant difference in Foxp3[+] T-cells, which were highest in the H99-infected brain and lowest in the Zc15-infected brain (Fig. 6D). We did not observe any significant differences in the numbers of lymphocytes, or CD4 T-cell polarization, in the lungs between any of the *C. neoformans* strains following intravenous infection (Fig. 6E and F). Taken together, these data demonstrate that CD4 T-cell recruitment to the infected brain is driven by fungal brain burden.

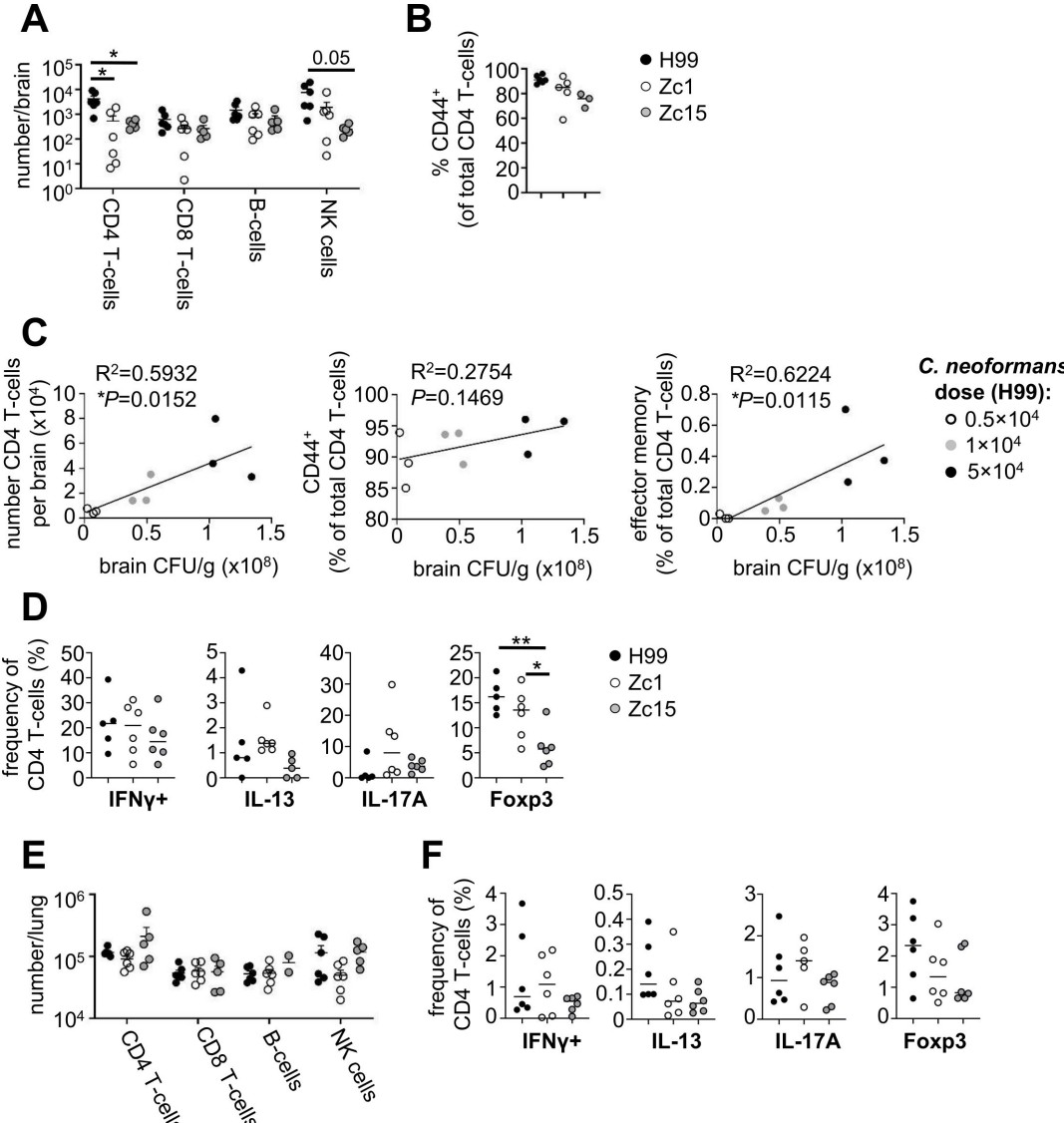

FIG 6  CD4 T-cell recruitment correlates with fungal brain burden. (A) Total numbers of indicated lymphocyte populations were quantified using flow cytometry (*n* = 6 mice per group). Data is pooled from two independent experiments and analyzed by two-way ANOVA. *$P$ < 0.05. (B) Frequency of CD44+ expression within the CD4 T-cell population in the brain in mice intravenously infected with *C. neoformans* H99, Zc1, and Zc15 at day 7 post-infection. (C) Mice were infected with different doses of *C. neoformans* H99 (*n* = 3 mice per dose), and indicated CD4 T-cell populations were quantified using flow cytometry and correlated with fungal brain burden (matched to the same animal). Data analyzed by simple linear regression. In all graphs, each point represents an individual animal. (D) Frequency of CD4 T-cells that were positive for indicated cytokines or Foxp3 at day 7 post-infection, in the brains of mice infected with H99, Zc1, or Zc15 (*n* = 6 mice per group). Data are pooled from two independent experiments and analyzed using one-way ANOVA. (E) Total number of indicated lymphoid populations in the lung of mice intravenously infected with *C. neoformans* H99 (*n* = 6 mice), Zc1 (*n* = 5–6 mice) and Zc15 (*n* = 5–6 mice). Data pooled from two independent experiments and analyzed by two-way ANOVA. (F) Frequency of CD4 T-cells that were positive for indicated cytokines or Foxp3 at day 7 post-infection in the lungs of mice infected with H99, Zc1, or Zc15 (*n* = 6 mice per group). Data are pooled from two independent experiments and analyzed using one-way ANOVA. *$P$ < 0.05, **$P$ < 0.01, ***$P$ < 0.005.

## *C. neoformans* Zc1 infection induces greater recruitment and TCR signaling by fungal-specific CD4 T-cells

Although we found a positive correlation between fungal brain burden and CD4 T-cell numbers in the brain, these experiments did not examine the antigen specificity of recruited CD4 T-cells. We therefore set out to determine whether recruitment of antigen-specific CD4 T-cell populations was similarly driven by fungal brain burden. For that, we adoptively transferred CnT.II TCR transgenic CD4 T-cells into mice prior

to infection (Fig. 7A). CnT.II T-cells express a TCR that specifically recognizes Cda2, an immunodominant antigen for this fungus (26). CnT.II donor animals were backcrossed onto *Nr4a3*-Tocky-IFNg-YFP reporter mice (27) to allow tracking of IFNγ production (YFP reporter) and TCR signaling (*Nr4a3*-Tocky reporter) by these cells *in vivo*. Using this model, we first measured the frequency of total CD4 T-cells that were antigen-specific in the brain (i.e., the proportion of total CD4 T-cells that were CnT.II transgenic cells). Surprisingly, we found a significantly higher proportion of CnT.II cells in the Zc1-infected brain than in H99- and Zc15-infected brains (Fig. 7B). There was less recruitment of CnT.II cells in the Zc15-infected brains compared with H99, although this did not reach statistical significance (Fig. 7B). Since Zc15 infection resulted in poor recruitment of CnT.II cells to the brain, we focused on H99 and Zc1 for further functional analysis. We found similar production of IFNγ, CD44 expression, and rates of cell division by CnT.II cells between H99 and Zc1 brain infection (Fig. 7C through E). However, TCR signaling, measured by expression of the *Nr4a3*-Tocky fluorescent reporter, was significantly higher in CnT.II cells infiltrating the Zc1-infected brain compared with the H99-infected brain (Fig. 7F). Collectively, these results show that while analysis of total CD4 T-cell populations may broadly mirror fungal brain burdens, specific analysis of fungal-specific CD4 T-cell responses is influenced by *C. neoformans* strain background that is partially independent of fungal burden.

## *C. neoformans* Zc1 boosts expression of MHCII by brain border macrophages

Since infection with *C. neoformans* Zc1 resulted in greater TCR signaling by fungal-specific CD4 T-cells in the brain compared with H99, despite lower fungal brain burdens and reduced total CD4 T-cell recruitment, we next explored how Zc1 infection influenced the expression of MHCII by brain-resident macrophage populations. Microglia are the most numerous brain-resident macrophage population but have poor capacity for antigen presentation and low expression of MHCII. In contrast, CD206$^+$ border macrophages are important antigen-presenting cells in the brain, co-expressing MHCII and positioned in sites, such as choroid plexus and meninges, to maximize antigen capture (28). CD206$^-$ macrophage populations are also found in the brain and are thought to be involved with tissue support functions due to their enhanced expression of trophic factors and reduced expression of MHCII (28). We infected mice with either H99 or Zc1 and then quantified the proportion of these resident macrophage populations that expressed MHCII by flow cytometry. This analysis revealed that the frequency of MHCII$^+$ microglia was significantly lower in the Zc1-infected brain compared to H99 (Fig. 7G and H). Total numbers of CD206$^+$ and CD206$^-$ border macrophages were significantly reduced in the Zc1-infected brain (Fig. 7I), in line with our earlier observations of reduced numbers of total brain macrophages with this *C. neoformans* strain. However, we found significantly increased frequency and expression of MHCII by both these populations in the Zc1-infected brain compared to H99 (Fig. 7J and K). We therefore hypothesized that the reduced fungal brain burden observed with Zc1 may be influenced by exaggerated fungal-specific CD4 T-cell responses, in part driven by enhanced MHCII expression on border macrophages. To examine that, we infected wild-type and *Rag2*$^{-/-}$ mice with either H99 and Zc1 and measured brain fungal burden. *Rag2*$^{-/-}$ mice lack lymphocytes and would therefore not exhibit the difference in T-cell responses between the two strains. However, we found a similar brain fungal burden between H99 and Zc1 in both the wild-type and *Rag2*$^{-/-}$ background (Fig. 7L). We also measured fungal burdens in the brains of wild-type and *Rag2*$^{-/-}$ mice infected with Zc1 at a later time point post-infection (day 10), where the effects of T-cells may be more apparent. However, fungal brain burdens at this time point also remained similar between groups (Fig. 7L). We were not able to perform a similar experiment at day 10 post-infection with H99, due to its higher virulence in this infection model (Fig. 1A). Taken together, these data indicate that our observations of enhanced MHCII expression by border macrophages and correlating increased fungal-specific CD4 T-cell responses that occur during infection with *C. neoformans* Zc1 do not have an overall impact on fungal infection control in the brain.

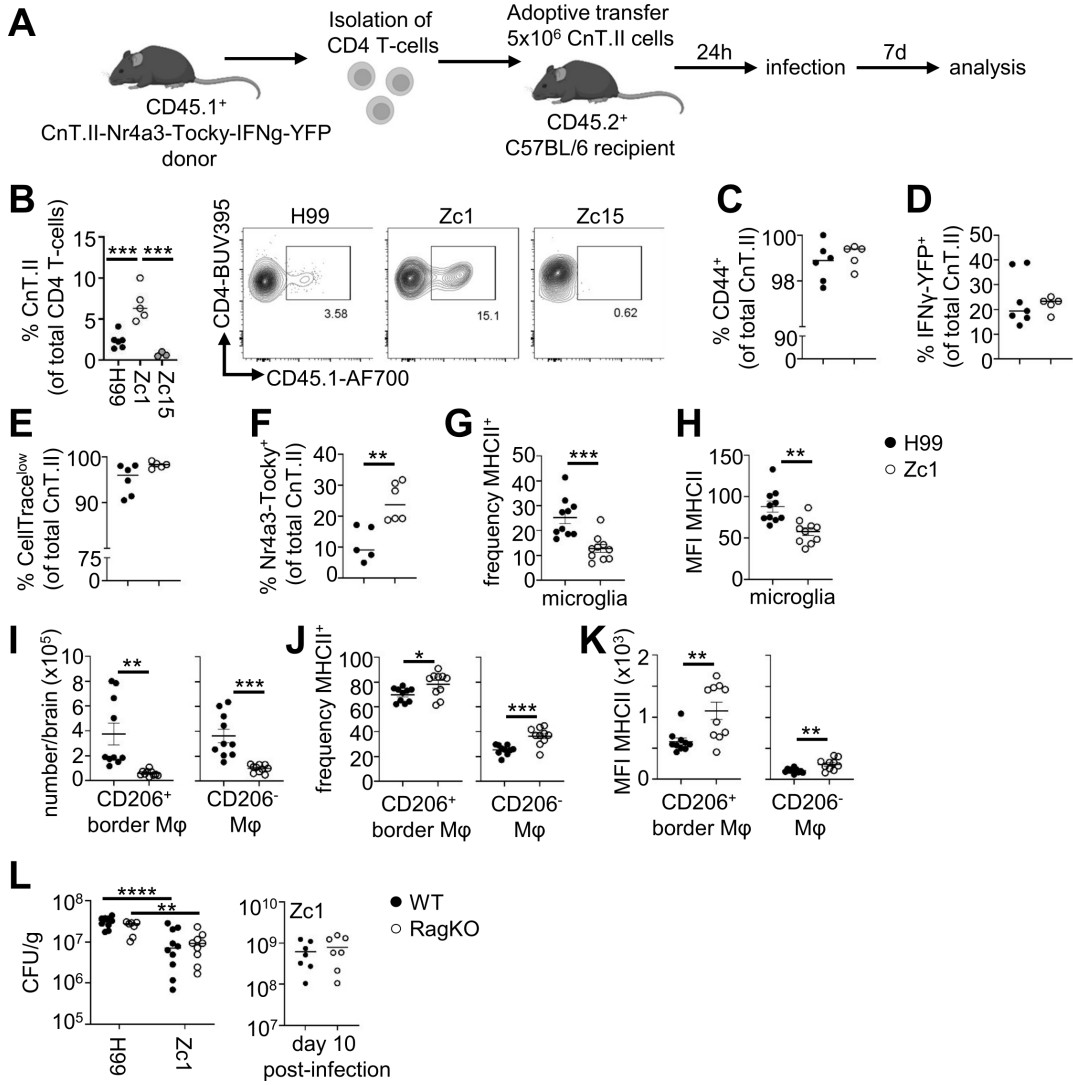

**FIG 7** Enhanced fungal-specific CD4 T-cell infiltration and MHCII+ border macrophages with *C. neoformans* Zc1 brain infection. (A) Schematic of adoptive transfer model used to track and analyze fungal-specific CD4 T-cell responses in the infected brain. (B) Frequency of fungal-specific CD4 T-cells (CnT.II) in the brain of mice infected with *C. neoformans* H99 ($n = 6$ mice), Zc1 ($n = 5$ mice), and Zc15 ($n = 5$ mice). Data pooled from two independent experiments and analyzed by one-way ANOVA. ***$P < 0.005$. Example FACS plots show CnT.II (CD45.1+) population within live CD4 T-cells. (C) Expression of CD44, (D) IFNγ-YFP, (E) retention of CellTrace proliferation dye and (F) expression of Nr4a3-Tocky by brain infiltrating CnT.II cells at day 7 post-infection in mice infected with *C. neoformans* H99 ($n = 6$–7 mice) or Zc1 ($n = 5$–6 mice). Data pooled from two independent experiments and analyzed by unpaired *t*-tests. **$P < 0.01$. (G) Frequency and (H) mean fluorescence intensity (MFI) of MHCII by microglia in brains of mice infected with *C. neoformans* H99 ($n = 10$ mice) and Zc1 ($n = 10$ mice) at day 7 post-infection. (I) Total number of CD206+ and CD206- macrophages, and (J) their frequency and (K) MFI of MHCII expression in brains of mice infected with *C. neoformans* H99 ($n = 10$ mice) and Zc1 ($n = 10$ mice) at day 7 post-infection. Data in panels G–K pooled from two independent experiments and analyzed by unpaired *t*-tests. *$P < 0.05$, **$P < 0.01$, ***$P < 0.005$. (L) Wild-type ($n = 10$ mice per group) and *Rag2*-/- ($n = 7$–9 mice per group) mice were infected with either *C. neoformans* H99 or Zc1 and brain fungal burdens analyzed at day 7 post-infection. Wild-type and *Rag2*-/- mice infected with Zc1 ($n = 7$ per group) were additionally analyzed at day 10 post-infection. Data pooled from two independent experiments were analyzed by Mann-Whitney U-tests. **$P < 0.01$, ****$P < 0.0001$.

## DISCUSSION

Several studies have compared mouse survival and fungal burdens following infection with different clinical isolates of *C. neoformans*, correlating these data with *in vitro* characteristics associated with virulence such as capsule size, melanin production, and stress responses (13, 15). However, many studies do not contain in-depth immune phenotyping of host leukocytes and/or do not analyze how organ-specific immune mechanisms may differ between strains. Our data demonstrate that organ fungal burden

and virulence of *C. neoformans* strains do not always equate to predictable immune responses, and this may be different depending on the organ and/or immune cell subsets analyzed. This is important to consider if the goal is to study immune cell behavior, particularly of tissue-resident cells that are the main drivers of organ-specific immune responses. We require more studies into host-fungal interactions during *C. neoformans* infection *in vivo*, especially with tissue-resident cells that are difficult to model *in vitro* and exhibit organ-specific and often subset-specific roles during this infection (11, 22). The data in this manuscript provide a framework to help investigators select strains and conditions to model an immune response of interest using the intravenous infection route, which allows for comparison of organ-specific immune responses while keeping variation in fungal burdens between animals to a minimum.

In our study, we used C57BL/6J mice, one of the most commonly used inbred mouse strains. We selected this strain because several knock-out and transgenic reporter animals are bred onto this background, including the antifungal TCR transgenic T-cell line CnT.II which we used to track fungal-specific CD4 T-cell responses in the brain (26). However, mouse strain is an important consideration in further analysis of organ-specific immune responses during *C. neoformans* infection, since recent studies have indicated significant differences in tissue-resident macrophage phenotype and function between inbred strains. For example, in the pleural cavity, there are two major subsets of tissue-resident macrophages that derive from monocytes (29). The differentiation of monocytes into these tissue-resident subsets can be expedited by recruited type-2 polarized CD4 T-cells during parasitic infection, which is required for protective immunity (29). However, while this protective pathway occurred in C57BL/6 mice, the conversion of monocytes to tissue-resident macrophages was stunted in BALB/c mice, helping explain why the BALB/c strain exhibited susceptibility to the parasite infection (29). Notable differences in susceptibility to cryptococcal infection between C57BL/6 and other inbred strains have now been observed by multiple groups, with recent work demonstrating a unique resistance phenotype in the SJL/J background (30). In those mice, resistance was attributed to the enhanced ability to mount protective type-1 (IFNγ-biased) immune responses to *C. neoformans* during infection (30). Indeed, prior work demonstrated a superior capacity of SJL/J-derived macrophages to produce IL-12p40, a type-1 polarizing cytokine (31). However, it is not known whether conversion of monocytes into different types of inflammatory and/or resident macrophages (e.g., lung interstitial subsets) depends on CD4 T-cell functional phenotype during *C. neoformans* infection, as was recently shown for pleural cavity macrophages (29). It is also not known whether macrophages have different functional capacities or variations in ontogenic lineages in SJL/J mice, which may also contribute towards their observed resistance to *C. neoformans* infection. It will therefore be important for future studies to determine whether potential differences in susceptibility between the inbred mouse strains are influenced by the dynamics of organ-resident immune cells and their antifungal responses. While our study focused on the influence of fungal strain, it may be important to determine how the immunology of inbred mouse strains influences control of infection in different organs, which will be required in future studies to fully examine the dynamics of host-fungal interactions.

Comparisons of clinical *C. neoformans* strains typically focus on expression and/or activation of well-known virulence traits for this fungus. This includes the formation of the capsule, which can shield the fungus against macrophage-mediated uptake and innate recognition, and production of melanin and urease, both of which can aid in resistance to oxidative killing and brain invasion, respectively (32). Another important aspect to *C. neoformans* virulence is morphology, and particularly the formation of Titan cells. These are large polyploid cells that generate in response to various stresses and are found within the infected lung (33). Titan cells are resistant to phagocytosis and can produce smaller daughter cells that may contribute to dissemination to the CNS. In addition, "seed cells" are another recently identified *C. neoformans* morphology that were shown to have enhanced capacity to invade the CNS and cause disease (34). Initial

comparisons of clinical *C. neoformans* isolates indicate that there is wide variability in the formation of Titan and seed cells under lab-tested conditions (17, 34). In our study, we used H99 and Zc1, which were both previously compared in Titan cell formation and related virulence potential in an inhalational model of the infection (17). Zc1 was found to produce smaller Titan cells, and similar to our findings, was a less virulent strain than H99 as there was less dissemination to the brain from the lung during infection (17), observations that were reproducible in our study. Titan cells are thought to be more relevant in lung infection where they have been observed within infected lung tissue in various studies, whereas they appear to be a rare occurrence within the brain. This is likely related to their large size that could prohibit their growth and/or dissemination within the narrow vasculature of the CNS (35). An important future direction for the field will be to determine how different *C. neoformans* morphologies alter uptake and/or subsequent activation of tissue-resident immune cells, which will depend on an agreement on how to define the different morphologies for this fungus.

In addition to physical variations between *C. neoformans* strains, a further important consideration that may influence outcome of immunology studies is the ability of *C. neoformans* to reside within macrophages. Following uptake by macrophages, *C. neoformans* may modify the intracellular environment and prevent fungal killing pathways, aiding its survival and immune evasion. This can occur via neutralization of phagosomal pH (36), or by secretion of CLP-1 which drives arginase expression within macrophages and blocks iNOS-mediated production of nitric oxide, which is toxic for the fungal cell (22). Interestingly, some macrophage subtypes appear to be more susceptible to intracellular *C. neoformans* infection than others, with lung interstitial macrophages and brain-resident microglia exhibiting higher rates of intracellular residence than other types of macrophages or recruited inflammatory cells, such as neutrophils (11, 22). Many of these mechanisms were defined using *C. neoformans* strain H99, which has a high propensity to grow within macrophages and develop an intracellular lifestyle in the host. However, whether these lessons are widely applicable to other *C. neoformans* strains is unclear, and the relative frequency of intracellular infection in human brain and/or lung is difficult to assess, given that autopsy samples only provide snap-shots. Indeed, these types of differences in the propensity to grow within host immune cells may lead to differences in how immunology studies are interpreted. For example, we recently found that depletion of microglia from adult mouse brain resulted in a reduction in fungal brain burden when mice were infected with H99 (11). This was due to the growth of the fungus within microglia, which represented an important intracellular niche for *C. neoformans* H99 in the brain. However, we did not find a difference in brain fungal burden in mice infected with Zc15 (11). For different *C. neoformans* strains, the lack of a protective role for microglia was shared, but the relative importance of microglia as an intracellular growth niche appeared to be more limited to H99 (11). This could reflect differences in the ability of the strains to modify phagosomal environments following uptake and/or differences in phagocytosis rates by myeloid cells, related to differences in cell wall/capsule and/or morphology, as outlined above. Regardless, this example demonstrates why confirmation of key immunology observations with multiple *C. neoformans* strains will be necessary in future studies going forward, particularly as the data in the current study demonstrate that while organ-specific immune responses may be heavily influenced by fungal strain, this is not always explained by differences in virulence.

Indeed, we found strain-dependent effects on fungal-specific CD4 T-cell responses that did not correlate with fungal brain burden. Infection with *C. neoformans* Zc1 resulted in greater recruitment of fungal-specific CD4 T-cells that had higher rates of TCR engagement which aligned with higher MHCII expression by border macrophages during infection. We recovered less viable yeast from Zc1-infected brains than H99, which we hypothesized was related to this enhanced T-cell response. However, infection of mice that lacked an adaptive immune system demonstrated that these differences in adaptive immunity did not contribute towards pathogen control, at least in this infection

model. However, this may have been affected by the time point we analyzed, which may preclude significant input from adaptive immunity. We used TCR transgenic CD4 T-cells that respond to the fungal antigen Cda2 to track and analyze fungal-specific responses in the brain. Cda2 is a chitin deacetylase that was discovered to be an immunodominant antigen in mice infected with *C. neoformans* (37) and is a protective antigen in vaccination studies (38). Mice immunized with Cda2-loaded glucan particles were significantly protected against lethal infection, which was dependent on the presence of CD4 T-cells but not B-cells or CD8 T-cells (39). Patients with cryptococcosis (who are T-cell sufficient) also have greater responses to Cda2 compared with healthy controls, indicating this may be an immunodominant antigen in humans too (40). Whether Cda2 activity and function differ between *C. neoformans* isolates is not known, however, our data suggest that this could be an important driver of variation in antigen-specific CD4 T-cell responses in experimental animal models. Future studies should aim to determine how *C. neoformans* strain variation may affect expression of immunodominant antigens and subsequent CD4 T-cell immunity, which may be an important determinant of vaccine responses and protection.

In summary, our data show important variation between organ-specific immune responses to different *C. neoformans* clinical isolates in an experimental animal model of acute disseminated infection, which is not always aligned to differences in fungal virulence and organ burden. It will be important to consider *C. neoformans* strain variation in immunology studies going forward, confirming key phenotypes with multiple strains to better align research findings between research groups and models.

## MATERIALS AND METHODS

### *C. neoformans* strains and infections

*C. neoformans* strains used in this study were H99 (serotype A, molecular type VNI), Zc1 (serotype A, molecular type VNI), and Zc15 (serotype A, molecular type VNB). Yeast was routinely grown in YPD broth (2% peptone [Fisher Scientific], 2% glucose [Fisher Scientific], and 1% yeast extract [Sigma]) at 30°C for 24 h at 200 rpm. For infections, yeast cells were washed twice in sterile PBS, counted using a hemocytometer, and yeast injected intravenously into the lateral tail vein. In general, mice were infected with $2 \times 10^4$ yeast, but in some experiments, the inoculum was changed to measure differences between doses (see figure legends for details of specific dose used). In some experiments, mice were infected with $2 \times 10^5$ yeast cells intranasally under isoflurane anesthesia.

### Mice

Eight- to 12-week-old female C57BL/6JCrl mice were purchased from Charles River and housed in individually ventilated cages under specific pathogen free conditions at the Biomedical Services Unit at the University of Birmingham, and had access to standard chow and drinking water *ad libitum*. Mice were euthanized by cervical dislocation at indicated analysis time points, or when humane endpoints (e.g., 18% weight loss, hypothermia, meningitis) had been reached, whichever occurred earlier. Mice were perfused with 10–20mL sterile PBS prior to isolation of organs.

### Determination of organ fungal burdens

For analysis of organ fungal burdens, animals were euthanized and organs weighed, homogenized in PBS, and serially diluted before plating onto YPD agar supplemented with penicillin/streptomycin (Invitrogen). Colonies were counted after incubation at 37°C for 48 h.

## Histology

Brains, lungs, spleens, and livers were dissected from mice at day 7 post-infection and placed in 10% formalin for 24 h. Fixed organs were then moved to 70% ethanol for storage until embedding in paraffin and sectioning. Tissue sections were stained with periodic-acid Schiff (PAS) (Merck) as per the manufacturer's instructions. Stained tissue sections were imaged using the Zeiss Axio Scan Z1 slide scanner. Final image analysis and measurements were completed using Zen Blue (v3.1) and Image J. A total of 50–70 regions of the lung sections were blindly evaluated for peribronchial infiltrates and alveolar involvement using a modified histological scoring system (41) originally described by Dubin et al. (24).

## Isolation of brain leukocytes

Leukocytes were isolated from brain using previously described in Materials and Methods (42). Briefly, brains were aseptically removed and stored in ice-cold FACS buffer (PBS + 0.5% BSA + 0.01% sodium azide) prior to smashing into a paste using a syringe plunger. The suspension was resuspended in 10mL 30% Percoll (GE Healthcare) and underlaid with 1.5 mL of 70% Percoll. Gradients were centrifuged at 2,450 rpm for 30 min at 4°C with the brake off. Leukocytes at the interphase were collected and washed in FACS buffer prior to labeling with fluorophore-conjugated antibodies and flow cytometry analysis.

## Isolation of lung leukocytes

Lungs were aseptically removed and placed in 4 mL digest buffer (RPMI, 10% FBS, 1% Pen/Strep, 1 mg/mL collagenase, 1 mg/mL dispase, and 4 0µg/mL DNase). The lungs were incubated in a water bath for 40–60 min at 37°C with intermittent shaking every 5–10 min. Lung tissue was then gently smashed using a syringe plunger and filtered through 100-µm filter. Lung cells were then collected by centrifugation at 1,400 rpm for 7 min at 4°C. Red blood cells were lysed on ice using PharmLyse solution (BD). Samples were washed in 2mM EDTA/PBS, filtered through at 40-µm filter into a fresh tube, and collected by spinning at 1,400 rpm for 7 min. Cells were resuspended in 200 µL FACS buffer, stored on ice, and placed in a FACS tube ready for staining.

## Flow cytometry

Isolated leukocytes were resuspended in PBS and stained with Live/Dead stain (Invitrogen) on ice as per manufacturer's instructions. Fc receptors were blocked with anti-CD16/32, and staining with fluorochrome-labeled antibodies was performed on ice. Labeled samples were acquired immediately or fixed in 2% paraformaldehyde prior to acquisition. In some experiments, samples were fixed and permeabilized using the Foxp3 staining buffer kit (eBioscience) prior to staining for intracellular antigens, performed overnight at 4°C. Anti-mouse antibodies used in this study were CD45 (30-F11), CD11b (M1/70), CX3CR1 (SA011F11), MHC Class II (M5/114.15.2), F480 (BM8), Ly6G (1A8), Ly6C (HK1.4), CD44 (IM7), CD69 (H1.2F3), CD127 (A7R34), CD45.2 (A20), CD45.1 (104), SiglecF (S1007L), CD64 (X54-5/7.1), IFNγ (XMG1.2), IL-17A (TC11-18H10.1), IL-13 (W17010B), Foxp3 (MF-14) all Biolegend, and CD4 (RM4.5), CD62L (MEL-14) from BD Biosciences, and MerTK (D55MMER) from eBioscience. Samples were acquired on a BD LSR Fortessa equipped with BD FACSDiva v9.0 software. Analysis was performed using FlowJo (v10.6.1, TreeStar).

## Adoptive transfers

CD4 T-cells were isolated from lymph nodes and spleens of CnT.II or CnT.II-Nr4a3-Tocky mice using magnetic beads-based CD4 T-cells isolation kit (Miltenyi Biotec) following the manufacturer's instruction. Purified CD4 T-cells were stained with 10 µM cell proliferation dye CFSE (Biolegend) or CellTrace eFluor450 (eBioscience) according to manufacturer's

instructions. A total of $5 \times 10^6$ CD4 T-cells were injected into recipient mice intraperitoneally. Mice were then infected with *C. neoformans* as above approximately 24 h after adoptive transfer. Mice were sacrificed on day 7 post-infection, and brains and lungs were dissected for further analysis by flow cytometry as above.

## Statistics

Statistical analyses were performed using GraphPad Prism 10.0 software. Details of individual tests are included in the figure legends. In general, data were tested for normal distribution by Kolmogorov-Smirnov normality test and analyzed accordingly by unpaired two-tailed *t*-test or Mann-Whitney *U*-test. In cases where multiple data sets were analyzed, two-way ANOVA was used with Bonferroni correction. In all cases, *P* values <0.05 were considered significant.

## ACKNOWLEDGMENTS

We thank Karen Bosworth and Claire Lyons and all staff at the Biomedical Services Unit for their help with animal care and husbandry. We thank Dr. Ferdus Sheik and support staff at the University of Birmingham Flow Cytometry Facility (RRID:SCR_027107) and Microscopy Facility (RRID:SCR_027108) for providing access to equipment and technical expertise with sorting, flow cytometry, and microscopy experiments.

This work was funded by the Academy of Medical Sciences (SBF004_1008, awarded to R.A.D.), the Medical Research Council (MR/S024611 awarded to R.A.D. and MR/T029137 awarded to R.A.D. and K.K.), and the Lister Institute for Preventative Medicine (awarded to R.A.D.).

## AUTHOR AFFILIATIONS

[1]Institute of Immunology & Immunotherapy, University of Birmingham, Birmingham, United Kingdom
[2]Department of Microbiology, Mycology and Immunology, Tohoku University Graduate School of Medicine, Sendai, Japan

## AUTHOR ORCIDs

Rebecca A. Drummond http://orcid.org/0000-0001-5424-7074

## FUNDING

| Funder | Grant(s) | Author(s) |
| --- | --- | --- |
| Academy of Medical Sciences | SBF004_1008 | Rebecca A. Drummond |
| Medical Research Council | MR/S024611, MR/T029137 | Rebecca A. Drummond |
| Lister Institute of Preventive Medicine | | Rebecca A. Drummond |

## AUTHOR CONTRIBUTIONS

Man Shun Fu, Conceptualization, Data curation, Formal analysis, Investigation, Supervision, Visualization, Writing – original draft, Writing – review and editing | Lorna George, Data curation, Formal analysis, Investigation, Supervision | Daisy Harris-Bosancic, Data curation, Investigation | Tahrim Hussain, Data curation, Investigation | Erin Clipston, Data curation, Investigation | Pui Mun Emily Chan, Data curation, Investigation | Lozan Sheriff, Data curation, Formal analysis, Investigation, Supervision | David Lecky, Data curation, Formal analysis, Investigation, Supervision | Kazuyoshi Kawakami, Conceptualization, Funding acquisition, Supervision | Rebecca A. Drummond, Conceptualization, Data curation, Formal analysis, Funding acquisition, Supervision, Visualization, Writing – original draft, Writing – review and editing

## ETHICS APPROVAL

Animal studies were approved by the Animal Welfare and Ethical Review Board and UK Home Office under Project Licence PBE275C33 and PP7564605.

## ADDITIONAL FILES

The following material is available online.

### Supplemental Material

**Supplemental material (Spectrum02517-25-s0001.pdf).** Fig. S1 and S2.

### Open Peer Review

**PEER REVIEW HISTORY (review-history.pdf).** An accounting of the reviewer comments and feedback.

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
