## [Reviewer comments · Microbiology Spectrum]

Microbiology Spectrum

Organ-specific immune responses are strain-dependent in a mouse model of *Cryptococcus neoformans* brain infection

Man Shun Fu, Lorna George, Daisy Harris-Bosanic, Tahrim Hussain, Erin Clipston, Pui Mun Emily Chan, Lozan Sheriff, David Lecky, Kazuyoshi Kawakami, and Rebecca Drummond

Corresponding Author(s): Rebecca Drummond, University of Birmingham

Review Timeline:

Submission Date:	August 18, 2025
Editorial Decision:	September 6, 2025
Revision Received:	November 19, 2025
Accepted:	December 19, 2025

Editor: Agostinho Carvalho

Reviewer(s): Disclosure of reviewer identity is with reference to reviewer comments included in decision letter(s). The following individuals involved in review of your submission have agreed to reveal their identity: Floyd L. Wormley, Jr. (Reviewer #1)

Transaction Report:

DOI: <https://doi.org/10.1128/spectrum.02517-25>

Re: Spectrum02517-25 (**Organ-specific immune responses are strain-dependent in a mouse model of *Cryptococcus neoformans* brain infection**)

Dear Dr. Rebecca A Drummond:

Thank you for the privilege of reviewing your work. Below you will find my comments, instructions from the Spectrum editorial office, and the reviewer comments.

Revision Guidelines

Sincerely,
Agostinho Carvalho
Editor
Microbiology Spectrum

Reviewer #1 (Comments for the Author):

This reviewer believes that this manuscript is indeed technically sound and that the study is experimentally and methodologically rigorous.

Reviewer #2 (Comments for the Author):

This study compared 3 clinical isolates of *C. neoformans* in two models of intravenous and inhalational infection in C57BL6 mice. Two of these strains are recently isolated and compared to a widely published reference strain, H99.

The study provides valuable information on the organ-specific differences between these strains in pulmonary and systemic infection scenarios. The fungal burden drives some of these differences, but some are linked to proposed variation in host responses to the strains in the local tissue environments.

The study appears to be technically sound and uses appropriate statistics.

However, some of the findings represent "snapshots" that are not completely connected, and there are two questions that beg answers, which would help to make the story complete.

First, it is the T-cell polarization status, which ultimately determines the fate of the response (not just their number and activation). Where the cytokine profiles in T-cells different between organs and strains? These findings would help to explain why the fungal loads and pathologies were different in each case. These additional steps could be done easily using intracellular flow or PCR of sorted CD4 T-cells.

The second issue is the lack of effect of T-cell absence in each of the studied models in Figure 7, where Rag2^{-/-} and WT mice are compared. This could be because the fungal burdens were tested too early to see the effect of T-cells or because the responses were maladaptive and T-cells did not contribute to the control of fungal burden. The later time points would help to resolve the kinetic issue, while cytokine responses requested above could explain whether the response was more maladaptive or more protective.

The additional data would help to clarify these points.

Reviewer #1 (Comments for the Author):

This reviewer believes that this manuscript is indeed technically sound and that the study is experimentally and methodologically rigorous.

Response: We thank the reviewer for recognising the technical strengths of our work.

Reviewer #2 (Comments for the Author):

This study compared 3 clinical isolates of *C. neoformans* in two models of intravenous and inhalational infection in C57BL6 mice. Two of these strains are recently isolated and compared to a widely published reference strain, H99.

The study provides valuable information on the organ-specific differences between these strains in pulmonary and systemic infection scenarios. The fungal burden drives some of these differences, but some are linked to proposed variation in host responses to the strains in the local tissue environments.

The study appears to be technically sound and uses appropriate statistics. However, some of the findings represent "snapshots" that are not completely connected, and there are two questions that beg answers, which would help to make the story complete.

First, it is the T-cell polarization status, which ultimately determines the fate of the response (not just their number and activation). Where the cytokine profiles in T-cells different between organs and strains? These findings would help to explain why the fungal loads and pathologies were different in each case. These additional steps could be done easily using intracellular flow or PCR of sorted CD4 T-cells.

Response: We have included new data on T-cell polarisation in revised Fig 6. We used intracellular flow cytometry and measured the frequency of CD4 T-cells that were positive for IFN γ , IL-17, IL-13 and Foxp3 in the brain and lung. We found little difference between the strains in these parameters, except for Foxp3⁺ CD4 T-cells in the brain which were significantly altered, mirroring the differences in brain fungal burden.

The second issue is the lack of effect of T-cell absence in each of the studied models in Figure 7, where Rag2^{-/-} and WT mice are compared. This could be because the fungal burdens were tested too early to see the effect of T-cells or because the responses were maladaptive and T-cells did not contribute to the control of fungal burden. The later time points would help to resolve the kinetic issue, while cytokine responses requested above could explain whether the response was more maladaptive or more protective.

The additional data would help to clarify these points.

Response: We added new data showing fungal brain burden at day 10 post-infection in wild-type and Rag2^{-/-} mice infected with the Zc1 strain. We attempted to get

similar data for H99-infected mice, however the higher virulence of this strain precluded attempts to analyse later time points. We have also added a sentence into the discussion to highlight this caveat of our study.

Re: Spectrum02517-25R1 (**Organ-specific immune responses are strain-dependent in a mouse model of *Cryptococcus neoformans* brain infection**)

Dear Dr. Rebecca A Drummond:

Dear Rebecca,

Your manuscript has been accepted, and I am forwarding it to the ASM production staff for publication. Your paper will first be checked to make sure all elements meet the technical requirements. ASM staff will contact you if anything needs to be revised before copyediting and production can begin. Otherwise, you will be notified when your proofs are ready to be viewed.

Sincerely,
Agostinho Carvalho
Editor
Microbiology Spectrum

Reviewer #2 (Comments for the Author):

The authors addressed my concerns, and the manuscript has improved. It can be accepted in its present form